# Study on Strength and Stiffness of WC-Co-NiCr Graded Samples

**DOI:** 10.3390/ma12244166

**Published:** 2019-12-11

**Authors:** Leszek Czechowski

**Affiliations:** Department of Strength of Materials, Lodz University of Technology, 90-924 Lodz, Poland; leszek.czechowski@p.lodz.pl; Tel.: +48-42-631-22-15

**Keywords:** instrumental indentation techniques, bending tests, finite element method, functionally graded materials, detonation gun layer deposition

## Abstract

This work deals with the investigation of the strength and the stiffness of samples built of step-variable functionally graded material (FGM). The considered FGM samples consist of two main components: WC and NiCr with some content of Co as a conjunction of structure. The samples were fabricated on the basis of the detonation gun layer deposition method which is regarded as a novelty in the case of FGM production. The analyzed samples possess a finite number of layers with different varying fractions of each constituent across the wall. The basic tests of bending were conducted to assess the influence of used components in adequate proportions on the stiffness and the total strength in bending. In addition, to validate the numerical approach, simulations of samples under similar loads with truly reflected material distributions were carried out. The material properties of components were determined due to micro-nano-hardness by instrumental indentation techniques. The numerical calculations were performed with the use of the material characteristics for each material and with a consideration of large deflections. Furthermore, by means of an electron microscope, the composition of materials and distribution of chemical elements across the thickness of samples were examined. This paper reveals the experimental results of FGM samples manufactured by detonation gun layer deposition which allows the creation of layer by layer moderately thin-walled structures. It was shown that the indentation method of a determination of Young’s modulus gave higher values in comparison to values attained in the bending test. Moreover, it was stated that a modelling of FGM still requires the study of each layer separately to clearly predict the strength of the whole FGM structure.

## 1. Introduction

The concept of a functionally graded material (FGM) was first shown in 1984 by Niino, a researcher from Japan. In the following years, he and others were focused on the development of FGMs [1,2,3,4]. At present, it is well known that FGMs are regarded as novel composite materials and are widely used in many industries: aerospace and nuclear technology, and civil, optical, automotive, biomechanical, electronic and chemical engineering. It should be mentioned that these materials still constitute a challenge for engineers. With regard to their special properties, FGMs are considered as future intelligent materials can work in untypical environments. Many FGMs are characterized by a great number of advantages such as high-temperature resistance or considerable reduction in the appearance of thermal stresses, among others. In assumption, FGMs possess a gradual change in effective material properties and a spacious change of non-uniform microstructure maintaining whole continuity. From the point of a theoretical approach, functionally graded materials are mostly a mixture of two components (glass, ceramics, metal) that fluently change from one to the other. Looking at the most-known compositions as ceramics and metal, ceramics are usually responsible for resistance to high temperatures, but metal can endure most plastic deformations. In regards to the literature, for over thirty years many works were devoted to the theoretical analysis of the behavior of FGM structures under thermal and/or mechanical loads [5,6,7,8,9,10,11,12,13,14,15,16,17]. One of the first works devoted to a study on FGM structures was elaborated by Praveen and Reddy [18], who analyzed the behavior of the plate under static and dynamic loads. Since the concept of the functionally graded material was presented, different techniques of the FGMs manufacturing were provided to obtain the structure in desirable properties [19,20,21,22]. Authors of paper [23], based on other research, assessed the possibility of the production of FGMs in different compositions. Lastly, much research has come out that deals with the preparation of FGM samples to examine the correctness of applied methods and mechanical or thermal properties. As it was proven in a great number of papers that the accurate manufacturing of desirable FGMs of adequate shape and dimensions with regard to the components configuration may cause many serious problems. It results especially in strongly differing mechanical and thermal properties: melting point, Young modulus and yield stress, among others. Taking a look at the literature, some methods of FGM fabrication were shortly below mentioned. Liu et al. in reference [24] used wire electric discharge machining and electrical discharge machining to prepare the samples of Ni–Al_2_O_3_ functionally graded material. Horizontal centrifugal casting technique for FGM fabrication was used in reference [25]. Matejíceka et al. [26] studied tungsten-steel composites and functionally graded materials produced from tungsten and steel powders by hot pressing. In paper [27], a functionally graded WC-TiC-Al_2_O_3_-graphene micronano composite with different Al_2_O_3_/(Al_2_O_3_+TiC) weight ratios was analyzed. Morin et al. [28] investigated FGMs produced by spark plasma sintering. The paper in reference [29] concerns Ti_3_SiC_2_/SiC functionally graded materials prepared by hot-pressing sintering where fracture toughness of samples was examined. On other hand, based on a theoretical approach, the implementation of functionally graded properties in finite elements for plate was studied in reference [30].

Owing to the fact that the assumed technique of FGM manufacturing determines the quality od obtaining appropriate properties in structures, the present work shows preliminary results of the bending of samples manufactured by detonation gun layer deposition method (DGLDM). In a study, samples made of WC–Co and NiCr steel as a mixture of these materials in adequate proportions, separately or in a multilayer graded material, were manufactured and tested. In addition, microscopic examinations of material composition and indentation methods on the same samples were conducted. This paper reveals the possibility of manufacturing multilayer graded materials by using DGLDM with regard to applied constituents, that significantly differ in both mechanical and thermal material properties. Furthermore, the results of all experimental tests indicated some mechanical properties and behavior of these materials. It is demonstrated that it is possible to fabricate multilayer thin-walled FGM samples on the basis of DGLDM which previously was only applied for coating materials.

## 2. Preparation of Samples

By means of DGLDM [31,32] all samples considered in analysis were prepared by INNTEG company (Bytom, Poland). The method has been developed over fifty years of the twentieth century as a method of layers deposition. At present, this method is often applied to cladding by the use of different powders [31,32,33,34,35]. The coatings were deposited onto aluminum substrates. After manufacturing samples, the bases were removed. To manufacture adequate samples, the powder with dimensions of 5–40 µm was used. Before the cladding process was followed, the substrates had been ultrasonically cleaned by using acetone. The powder materials in some proportion depending on the deposited layer were inserted into the tank. Argon gas was employed to protect the powder. The single coating thickness amounted to from 50 µm to 100 µm. This technique of spraying enables, during processing, the motion of particles with a velocity above 340 m/s (ultrasonic velocity), or even 1000 m/s. The dimensions of manufactured samples couldn’t be unified with respect to the complexity of fabrication and amount to the following ranges: thickness of 1.3–2.35 mm, width of 19.1–20.0 mm and length of 112.2–125 mm. The types and dimensions of prepared samples are shown in Table 1.

## 3. Methods of Material Properties and Composition

### 3.1. Bending Test

All prepared samples were subjected to a three-point bending test on a stand equipped with an Instron machine. This machine ensures performing tests for compression and tension with a range of 0.2 to 200 kN. For smaller forces it can be possible to additionally apply the force gauge with a narrower scope. For the bending test, the grip set to the span distance of 80 mm was used. Radius of the tip indenter and the support was the same and equal to 5 mm. The tests of bending were executed based on the standard PN-EN ISO 7438:2016-03, in reference [36].

### 3.2. FE Analysis

Numerical simulations were provided on the basis of finite element method in Ansys 18.2^®^ version software [37]. By using an 8-node solid finite element 185, discrete models were elaborated as is shown in Figure 1. Across the thickness of a sample, the number of finite elements amounted to 20. The total number of finite elements for the studied samples was 7500. Half a model was modelled by imposing the symmetry conditions. Between the sample and the movable indenter at a radius of 5 mm or supports, contact and target elements were assumed (170/174). The simulation was done for large deflections by using Green–Lagrangian equations. Based on the Newton–Raphson algorithm, nonlinear computations conducted in steps were followed. The number of steps was set to be from 50 up to 500. The maximum number of possible iterations during a single step was increased from the default setting to 200. The support was fixed and the indenter was controlled by enforced displacement. Subsequently, the bending forces were taken from reaction forces at the support.

### 3.3. Microscopic Examinations

The microscopic examinations of both the material composition and the material distribution across the samples thickness were performed by means of field emission scanning electron microscope JSM–6610LV (JEOL, Tokyo, Japan). The microscopic examinations for the chosen samples were conducted in co-operation with the Institute of Material Science and Technology (Lodz University of Technology). The morphology and the material distribution of the studied samples were studied in ambient conditions throughout thickness in the vicinity of the gravity center of samples.

### 3.4. Indentation Method

By verifying some mechanical properties as Young’s modulus, a MTS NANO INSTRUMENTS model G200 setup was employed. This stand was composed of an anti-vibration table, nano-indenter, anti-vibration chamber with thermal/acoustic insulation, and a CSM control unit NanoSwift control utility. The measurements of indentation were carried out by using a Berkovich indenter about half an angle 65.3° and tip radius below 20 nm. These tests were executed in the Institute of Material Science and Technology (Lodz University of Technology). The forces of penetration applied to the tested surface amounted to 15–25 µN. The calibration of the tip shape was performed based on a fused silica standard. The obtained data were elaborated on the basis of an approach given in reference [38]. For each point, nine measurements with the strain rate of 0.05/s were made.

## 4. Results

### 4.1. Bending Test

A three-point bending test was conducted for 7 different samples included in Table 2. The study was aimed at an assessment of the strength and the stiffness of detailed samples. Because of a verification of preliminary examinations and continued high fabrication costs, only single samples at their determined composition and build were taken into consideration. Figure 2 presents the results of bending as maximum bending stress vs. deflection in the middle between supports. Brittle breaking and almost proportional deflection vs. load for all considered samples were observed until the sample broke. Slightly nonlinear behavior was noted for Sample 2. The maximum bending stress was achieved for Sample 4 (75% NiCr and 25% WC–Co, maximum bending stress 332.5 MPa) though among the considered samples the most strength should be assigned to Sample 1, made of NiCr (maximum bending stress amounted to 289.5 MPa). In the case of Sample 4, participation of constituent 1 (NiCr) was 3 times more than the presence of constituent 2 (WC–Co). Hence, this difference is not significant. The weakest sample turned out to be Sample 2 (made of 100% WC–Co). The maximum stress at the damage for this case was equal to 156.5 MPa. Thus, increasing the share of NiCr, the ultimate strength in bending grew as well because for Sample 3 (composed of constituent 1–50% and constituent 2–50%), the maximum stress was recorded as 213.9 MPa.

A reversed situation was observed if Young’s moduli were considered. In the case of the pure WC–Co, the calculated Young’s modulus came to 93 GPa. This value is almost 4 times smaller than the foreseen value but it was the greatest one among the other samples. In the case of Sample 1 (pure NiCr), the modulus of elasticity amounted to about 50 GPa. For 4-layer graded samples, the maximum stresses in bending and Young’s moduli were from 248.0 MPa to 292.0 MPa and from 59.5 GPa to 77.1 GPa, respectively. Those values were investigated in the middle of samples between boundaries. To highlight those values, all the characteristic mechanical properties were inserted into Table 2.

### 4.2. Validation of FE Model

By validating the stiffness of samples, numerical estimations on the basis of the finite element method were done (Figure 3). The computations were conducted on discrete models with almost the same dimensions as in the experiment (FEM—numerical results, EXP—empirical results). The target of this study was fitting material properties to attain similar (close) curves (in other words, to match the results of numerical models). The numerical calculations were performed for Samples 1 and 2 and Sample 5_1. Young’s moduli for numerical calculations were assumed as given in Table 3. Another parameter of Poisson’s ratios were assumed, based on reference [38] and the mixture law. As it was displayed, at these parameters the characteristics of bending possess comparable and close trends.

### 4.3. Microscopic Examinations

Some of the microscopic examinations were illustrated in Table 4, Table 5 and Table 6 (Sample 1, Sample 2 and Sample 5_1, respectively). The compositions of elements were investigated in the range from 11 to 23 points across the thickness in the middle of the samples after breaking. The elements of each considered sample for different zones and for 6 chosen points were illustrated. It was revealed that in the case of Sample 1, moderately equal and intended distributions of elements were obtained (see Table 4). In this case, the average content of elements amounted to: Ni—40.5%, C—35.2%, Cr—14.4% and others—9.9%.

In case of Sample 2, metallurgy examinations were performed for 11 points. The average content of elements amounted to: C—72.4%, W—18.3%, Co—8.5% and others—0.8%. For this technique of the sample manufacturing, dispersions of compositions amounted to about 10% of each chemical element. The total content of tungsten comes to one fifth of all elements in the structure, but this effect significantly decreased the strength of the samples in bending. However, in contrast it raises summary of Young’s modulus (see Figure 3). In the case of Sample 5_1 (FGM), four layers were examined. The content of the elements was gradually varying in a dependence of position. The view of the element distribution across the thickness of the sample taken from a microscope was composed of two pictures because it was difficult to examine all the points in one set (Table 6). In the middle layers, some existence of both components is seen (circa 50% of NiCr and circa 50% of WC–Co). In the highest layer sample, one can observe large amounts of of NiCr (WC–Co is lacking, as was expected). A reverse situation was observed in the lowest layer (points 92 and 94—content of pure tungsten is about 86.5% and 88%). For this layer, the composition in the structure excludes completely the presence of NiCr. Hence, based on this method of FGM fabrication, it can be concluded that noticeable diffusion of elements didn’t occur.

### 4.4. Indentation Results

Micro-indentation techniques for determining Young’s moduli for Samples 1, 2 and 5_1 were employed in order to compare with the experimental investigation. The measurements were conducted in the center of the samples throughout the thickness. In each measure point, 9 imprints were consecutively taken. All the values appeared to be considerably greater than those received in the bending tests.

In the case of Sample 1 (NiCr) Young’s modulus amounted to circa 200 GPa (Figure 4) and is very close to the predicted value. The results of elasticity modulus for Sample 2 (WC–Co) are presented in Figure 5. For this material the modulus averaged out at above 300 GPa (for some single measurements, even 400–600 GPa). In the case of Sample 5_1 (a sample composed of 4 layers from NiCr to WC–Co) the trend of Young’s modulus was increasing but at the end a slight drop was noticed, apparently due to a locally weaker structure (Figure 6). In general, those values related to indentation methods are meaningfully greater than on the basis of results achieved in bending tests. It is probably caused by a weaker connection of the full structure, or microscopically higher values of toughness.

## 5. Summary

In this paper, the mechanical properties of samples manufactured through a detonation gun layer deposition method were studied. In addition, two investigations on samples were performed: indentation measurements and composition examinations. Regarding the achieved results, it was mainly noted that:the employed method of manufacturing multi-layer thin-walled samples enables building a functionally graded material by the use of different components and elements;Young’s moduli measured during bending tests differ from those measured by the indentation technique;based on the obtained results, real material properties and distributions should be verified depending on method, obtained thicknesses of layers and composition;the method of indentation can be employed to give some results of elasticity but cannot be used as a basic test in determining total material properties;the paper includes preliminary investigation on thin-walled FGM samples from the point of a proposed technique as a detonation gun layer deposition method and applied constituents comprising a structure of FGM. It should be underlined that the present method can be further developed with consideration of other compositions and a greater number of layers.

## Figures and Tables

**Figure 1 materials-12-04166-f001:**
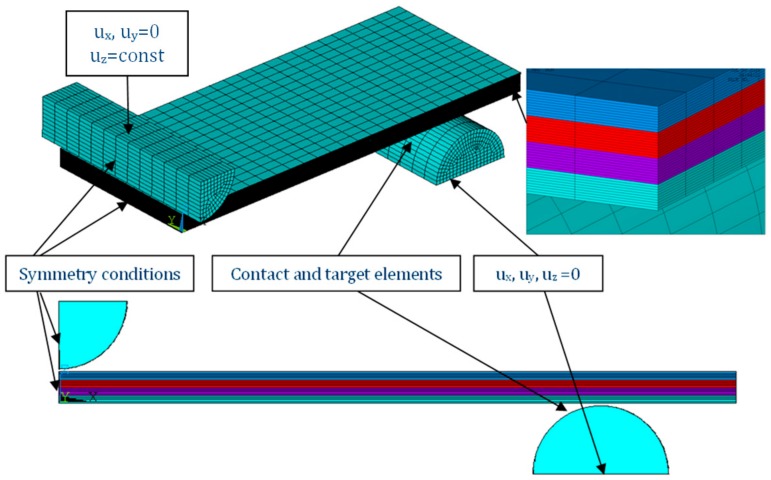
FE with boundary conditions.

**Figure 2 materials-12-04166-f002:**
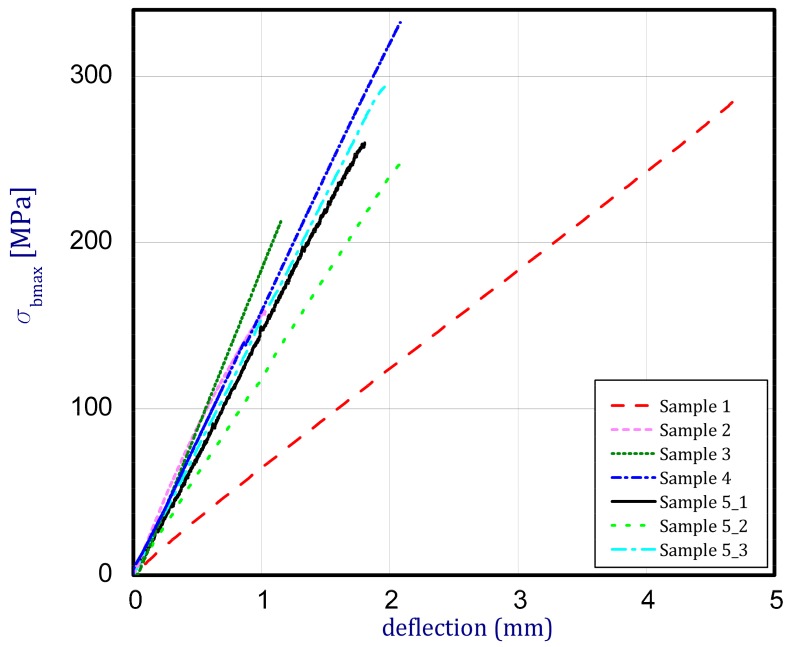
Center deflection of samples vs. bending stress (experimental results).

**Figure 3 materials-12-04166-f003:**
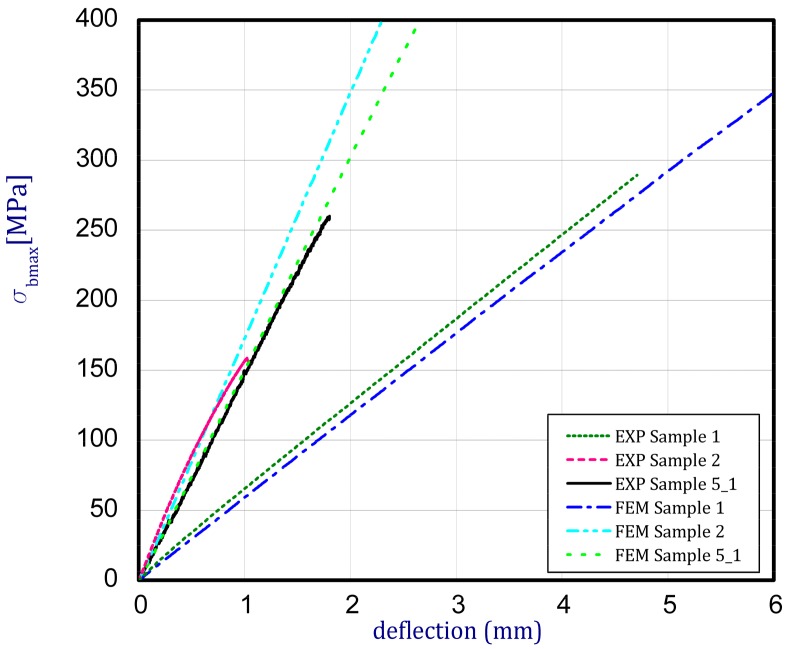
Center deflection of samples vs. bending stress.

**Figure 4 materials-12-04166-f004:**
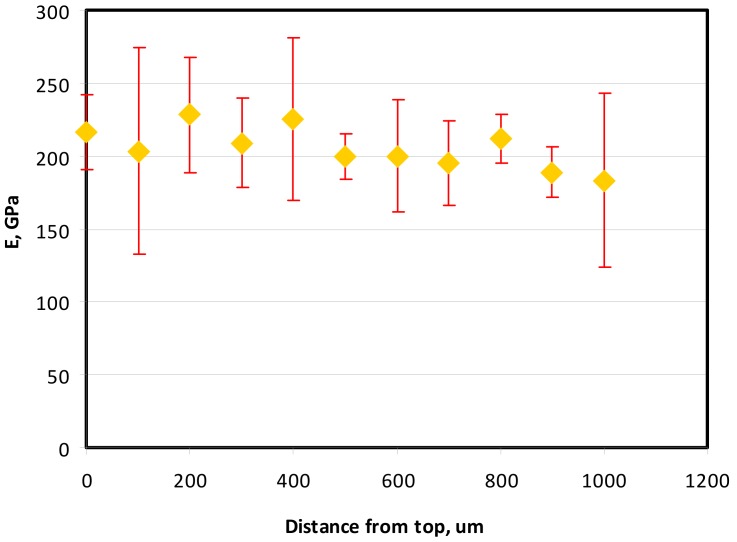
Young’s modulus for Sample 1.

**Figure 5 materials-12-04166-f005:**
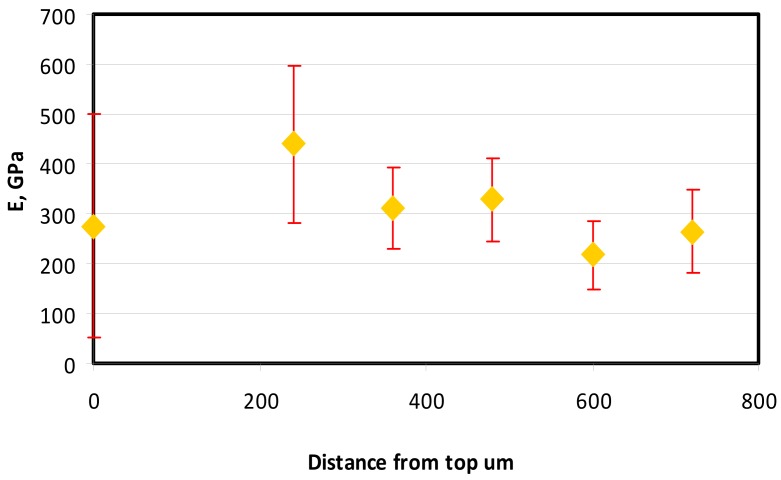
Young’s modulus for Sample 2.

**Figure 6 materials-12-04166-f006:**
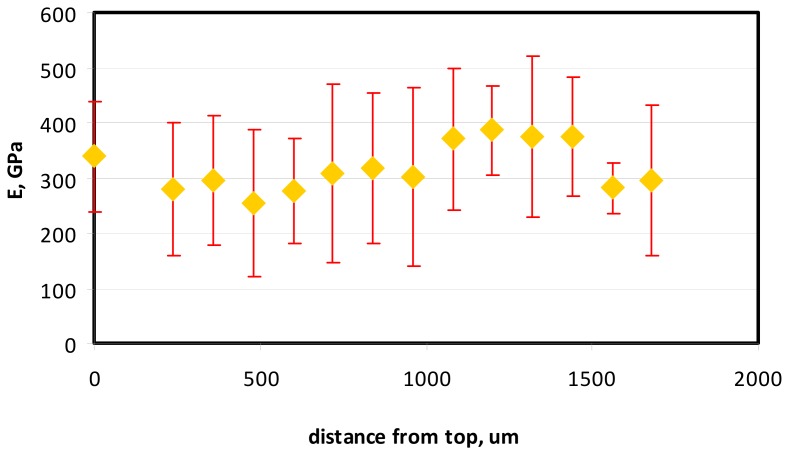
Young’s modulus for Sample 5_1.

**Table 1 materials-12-04166-t001:** Samples considered in study.

	Sample 1	Sample 2	Sample 3	Sample 4	Sample 5_1 (From Left) Sample 5_2 Sample 5_3
View of samples	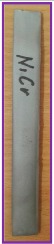	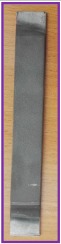	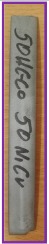	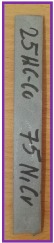	
Composition of main constituents	100% NiCr	100% WC–Co	50% NiCr 50% WC–Co	75% NiCr 25%WC-Co	Multilayer material (FGM): 1st layer—100% NiCr 2nd layer—25% WC–Co —75% NiCr 3rd layer—75% WC–Co —25% NiCr 4th layer—100% WC–Co
Dimensions: thickness (mm) width (mm) length (mm)	1.2 19.2 120	2.0 19.6 122	1.6 20.0 121	2.35 19.8 125	2.1–2.3 mm (each layer circa 0.525–0.575 mm) 19.1–20.0 112.2–120

**Table 2 materials-12-04166-t002:** Mechanical properties of analyzed samples due to bending tests.

Number of Sample	Maximum Bending Force (N)	Young’s Modulus (GPa)	Ultimate Strength in Bending (MPa)
Sample 1	66.7	51.2	289.5
Sample 2	103.8	93.0	156.5
Sample 3	150.5	75.0	213.9
Sample 4	302.4	70.0	332.6
Sample 5_1	230.2	72.0	260.1
Sample 5_2	172.7	59.5	248.0
Sample 5_3	219.3	77.1	292.0

**Table 3 materials-12-04166-t003:** Mechanical properties of analyzed samples used for numerical simulations.

Number of Sample	Young’s Modulus (GPa)	Poisson’s Ratio [39] (MPa)
Sample 1	50	0.3
Sample 2	90	0.25
Sample 5_1	1st layer—50	0.3
	2nd layer—70	0.27
	3rd layer—80	0.28
	4th layer—90	0.25

**Table 4 materials-12-04166-t004:** Composition of elements for Sample 1.

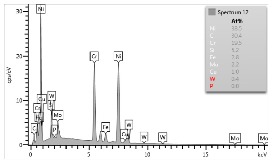	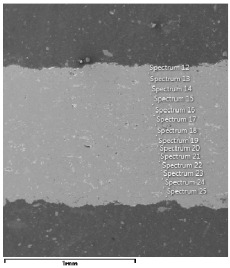 Average content of elements (from 14 points): Ni—40.5% C—35.2% Cr—14.4% Others—9.9%	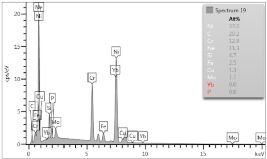
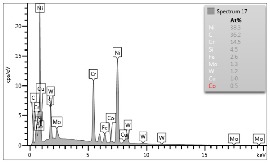		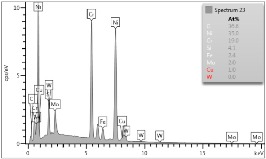
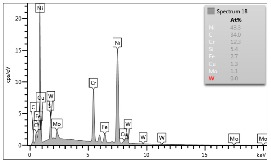		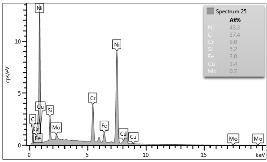

**Table 5 materials-12-04166-t005:** Composition of elements for Sample 2.

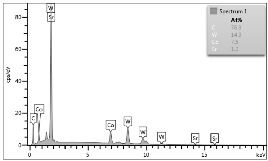	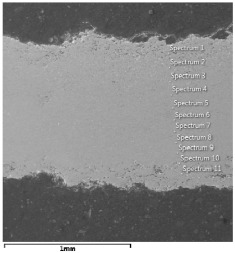 Average content of elements (from 11 points): C—72.4% W—18.3% Co—8.5% Others—0.8%	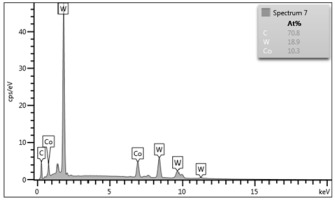
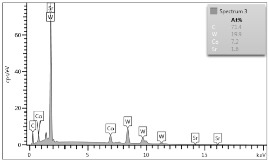		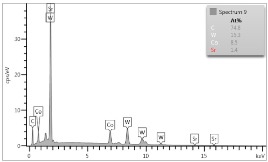
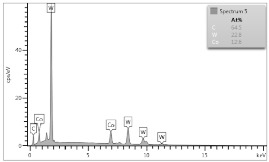		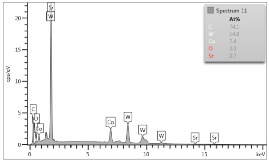

**Table 6 materials-12-04166-t006:** Composition of elements for Sample 5_1.

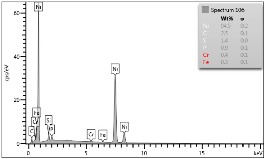	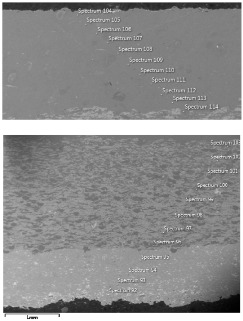 Content of elements (from 23 points): Ni—0–90.8% C—8–35.2% Cr—0.4–5% Others—9.9% C—72.4% W—0–89.3% Co—8.5% Others—0.8%	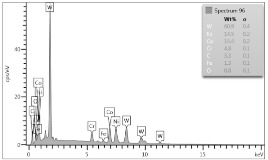
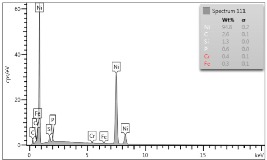		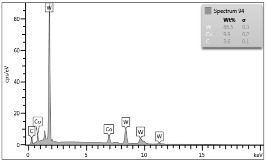
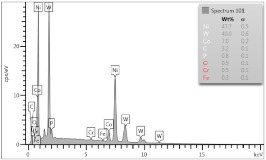		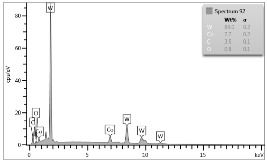

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
