# Peer review of "Study on Strength and Stiffness of WC-Co-NiCr Graded Samples"

_materials, 2019, doi:10.3390/ma12244166_

Round 1

Reviewer 1 Report

11:... the dotonation gun...

12:.... in the case....

15: .... the numerical....

20:.... of an electronic...

24:  in comparison....

31: ... a Researcher...32:  on the development....

38:  high-temperature

45 for over thirty...... works were....

79: The order of the references is flasch from line 79. Literature 30, line 82, is mentioned later on.

289: replace ; by .

294: Set the year bold.

Author Response

Dear Reviewer,

first of all, I’d like to thank for your valuable remarks, which let me refine my paper. Taking into consideration these suggestions, I have done my best to correct my article and to fulfil the requirements of publishable standard. The added or changed text was highlighted on “yellow”.

With best Regards,

L. Czechowski

Reviewer 2 Report

The author studies the strength and stiffness of functionally graded materials (FGMs). Specifically, the research is conducted on WC-Co-NiCr FGMs that are manufactured by means of detonation gun layer deposition.
The reviewer has mixed feelings with respect to this work. On the one hand, it has novelty; the work is more novel than the average paper published on this journal. However, the novelty lies on the manufacturing method and it appears that this has been done entirely by an external company.
The following issues have to be taken into consideration to make the paper suitable for publication:
1) Very few details are given regarding the manufacturing method, where the novelty of the work lies (not on the method itself, but the method applied to FGMs).
2) While some FGM manufacturing methods have been covered in the introduction, there are many missing.
3) The reviewer is a bit confused by the use of shell finite elements for this specific case study.
4) The assumption of step-wise variation of elastic properties is not clear. According to Table 1, the variation of the main constituents appears to be gradual and a rule of mixtures will provide with a continuously varying elastic profile. While the author is not expected to change the finite element model, the choice of “homogeneous” finite elements (versus graded finite elements) should be elaborated. And this should be done in the context of the literature; see, for example, in this journal: Materials, 12(2), 287 (2019)
5) It would be interesting to conduct further experimental studies, such as those involving fracture, as it has been done in other model FGMs. However, the reviewer understands that this may be out of the scope of the work.
6) The paper needs proof-reading. See, e.g., page 2: “the method has been developed from fiftieth years of twentieth century”

Author Response

(The authors gave the same response as above.)

Round 2

Reviewer 2 Report

The author has tried to clarify the comments and corrected some aspects of the paper. While further changes could have been ideal, there is undoubtedly novelty in the work and this reviewer recommends now publication.

Two comments are left for the author's consideration:

1) The statement "The generation of finite elements was obtained from shell elements" could be misleading. The reviewer now understands that it refers to the mesh generation but maybe these details could be removed.

2) The English is still not perfect, despite the effort. Another revision (maybe by professional services) will make sure that the quality of the text is a level according to the quality of the science.

Author Response

Dear Reviewer,

once again I’d like to thank for your valuable remarks  and suggestions.

Best Regards,

L. Czechowski

This manuscript is a resubmission of an earlier submission. The following is a list of the peer review reports and author responses from that submission.

Round 1

Reviewer 1 Report

In your manuscript you were trying to present mechanical properties of functionally graded WC-Co-NiCr materials. 

After reading your manuscript carefully, I have to reject its publication due to some serious flaws. General remark - English must be significantly improved. 

Already in the title you wrote "...WC-NiCr graded samples", but you prepared WC-Co - NiCr FGMs. 

Abstract does not include the most important findings of this research - have to be revised. 

Introduction has to be re-written in more concise way.

Experimental: the information about starting powders is missing.

Methods: Figs 1, 3 and 4 are not necessary (the researchers which are working in the field of interest know, how the used machines look like and their appearance is not relevant for the study). 

Results & Discussion 

Please check, how the results should be presented. Especially microscopy (and it is not microscopy measurements that you were performing, but microscopy analysis/examination etc...)

The discussion is missing!

Author Response

Dear Reviewer,

thank a lot for review of my manuscript.

The title of manuscript has been changed from

"Study on strength and stiffness of WC-NiCr graded samples"

to

"Study on strength and stiffness of WC-Co-NiCr graded samples".

L. Czechowski

Reviewer 2 Report

Hello.

I suggest you improve the quality of photographs of samples and equipment (for example, figure 4). Maybe this photo should be deleted?

figure 1 - intender - indenter?

figure 1 - specimen - it’s better to use everywhere - a sample

148 - simple - sample

206 - 300GPa, gap

Author Response

Dear Reviewer,

thank a lot for review of my manuscript.

The title of manuscript has been changed from

"Study on strength and stiffness of WC-NiCr graded samples"

to

"Study on strength and stiffness of WC-Co-NiCr graded samples"

L. Czechowski

Reviewer 3 Report

In my view, literature references are missing. This concerns the methods used, line 68 DGLDM, 77 INNTEG, and 103 PN-EN ISO. Further material values or data as in line 161 the intended value and in table 3 the Poisson ratios.

In line 142, reference is made to Table 3, but it should be Table 2. In line 145 "The" is not necessary in front of Fig. 5.
In Fig. 6, line 176, it should be explicitly pointed out that this is a numerical result. The same applies to column 2 in Table 3 (line 178). For section 4.4 there is no interpretation why the values deviate from the bending test.

Author Response

(The authors gave the same response as above.)

Round 2

Reviewer 1 Report

Although some of the comments were considered in this revised version, there is still issues that prevent me from recommending this paper for publication. 

English needs to be improved. i.e. lines 79 and 80.  Materials and methods: based on your description of this part, the work can not be reproduced.  Microscopy: please check how microscopy analyses should be presented. You did not even include the information which kind of fraction (%) are you presenting.  Discussion is missing. (Again!)